# Microbiota Composition of Breast Milk from Women of Different Ethnicity from the Manawatu—Wanganui Region of New Zealand

**DOI:** 10.3390/nu12061756

**Published:** 2020-06-11

**Authors:** Christine A. Butts, Gunaranjan Paturi, Paul Blatchford, Kerry L. Bentley-Hewitt, Duncan I. Hedderley, Sheridan Martell, Hannah Dinnan, Sarah L. Eady, Alison J. Wallace, Sarah Glyn-Jones, Frank Wiens, Bernd Stahl, Pramod Gopal

**Affiliations:** 1The New Zealand Institute for Plant and Food Research Limited, Private Bag 11600, Palmerston North 4442, New Zealand; paul.blatchford@zespri.com (P.B.); kerry.bentley-hewitt@plantandfood.co.nz (K.L.B.-H.); duncan.hedderley@plantandfood.co.nz (D.I.H.); sheridan.martell@plantandfood.co.nz (S.M.); hannah.dinnan@plantandfood.co.nz (H.D.); pramod.gopal@plantandfood.co.nz (P.G.); 2The New Zealand Institute for Plant and Food Research Limited, Private Bag 92169, Auckland 1142, New Zealand; gunaranjan.paturi@plantandfood.co.nz; 3Zespri International Limited, 400 Maunganui Road, Mt Maunganui 3149, New Zealand; 4The New Zealand Institute for Plant and Food Research Limited, Private Bag 4704, Christchurch 8140, New Zealand; sarah.eady@plantandfood.co.nz (S.L.E.); alison.wallace@plantandfood.co.nz (A.J.W.); 5Nutricia New Zealand, 56-58 Aintree Avenue, Mangere, Auckland 2022, New Zealand; sarah.glyn-jones@danone.com; 6Danone Nutricia Research, Uppsalalaan 12, 3584 CT Utrecht, The Netherlands; frank.wiens@danone.com (F.W.); bernd.stahl@danone.com (B.S.); 7Human Nutrition & Health, DSM Nutritional Products, P.O. Box 2676, CH-4002 Basel, Switzerland; 8Utrecht Institute for Pharmaceutical Sciences, Department of Chemical Biology & Drug Discovery, Utrecht University, Universiteitsweg 99, 3584 CG Utrecht, The Netherlands

**Keywords:** human milk, breastfeeding, ethnicity, microbiota, immune modulators

## Abstract

Human breastmilk components, the microbiota and immune modulatory proteins have vital roles in infant gut and immune development. In a population of breastfeeding women (*n* = 78) of different ethnicities (Asian, Māori and Pacific Island, New Zealand European) and their infants living in the Manawatu–Wanganui region of New Zealand, we examined the microbiota and immune modulatory proteins in the breast milk, and the fecal microbiota of mothers and infants. Breast milk and fecal samples were collected over a one-week period during the six to eight weeks postpartum. Breast milk microbiota differed between the ethnic groups. However, these differences had no influence on the infant’s gut microbiota composition. Based on the body mass index (BMI) classifications, the mother’s breast milk and fecal microbiota compositions were similar between normal, overweight and obese individuals, and their infant’s fecal microbiota composition also did not differ. The relative abundance of bacteria belonging to the *Bacteroidetes* phylum was higher in feces of infants born through vaginal delivery. However, the bacterial abundance of this phylum in the mother’s breast milk or feces was similar between women who delivered vaginally or by cesarean section. Several immune modulatory proteins including cytokines, growth factors, and immunoglobulin differed between the BMI and ethnicity groups. Transforming growth factor beta 1 and 2 (TGFβ1, TGFβ2) were present in higher concentrations in the milk from overweight mothers compared to those of normal weight. The TGFβ1 and soluble cluster of differentiation 14 (sCD14) concentrations were significantly higher in the breast milk from Māori and Pacific Island women compared with women from Asian and NZ European ethnicities. This study explores the relationship between ethnicity, body mass index, mode of baby delivery and the microbiota of infants and their mothers and their potential impact on infant health.

## 1. Introduction

The gut microbiota has a dynamic role in maintaining host health by preventing the colonization of pathogenic bacteria and maintaining mucosal immunity [1]. The initial microbial colonization of the infant gut may begin before birth [2], but is further established during the birth process and through various transmission routes from the mother and the environment [3,4]. One of the major determinants of infant gut microbiota composition initially is mode of birth, whereby babies delivered vaginally come into contact with the maternal bacteria through the maternal feces and birth canal, though other factors are important including birth environment, prematurity, hygiene, mother-infant contact and the method of infant feeding. Further, maternal metabolic health can influence the infant gut microbiota composition and its related health outcomes [5]. Breast milk has been found to contain several bacterial groups with *Bifidobacteriaceae*, *Streptococcaceae* and *Staphylococcaceae* families being members of the core breast milk microbiome [6,7]. The microbiomes in the mother’s breast milk and infant feces are closely related [8,9], emphasizing the route of transmission from mother’s milk to the infant gut. Moreover, the mother’s breast milk is important in shaping the infant gut microbiome for better health outcomes later in life.

In addition to the nutritional components, breast milk contains immune bioactive factors [10]. These include regulatory cytokines, growth factors, peptides, fatty acids and oligosaccharides and their presence is thought to support innate immunity and direct the development of adaptive immunity in the infant in the early weeks of life when breast milk is the sole source of nutrition [11,12,13]. Large quantities of secretory immunoglobulin (Ig) A are present in breast milk. The IgA antibodies are capable of neutralizing the pathogens prior to binding and infecting the infant cells [14]. The secretory IgA antibodies present in breast milk confer immunological protection to the infant during the first few months of life and are part of the establishment of long-term intestinal homeostasis by regulating gut microbiota composition and gene expression in the intestinal epithelial cells [15].

There is limited information on human breast milk composition with regard to ethnicity and the effect this may have on bacterial populations and immune modulatory proteins present in the milk. In the present study, microbiota composition and immune modulatory protein concentrations in breast milk from women of different ethnic groups living in New Zealand (NZ) were examined. In addition, fecal microbiota composition in the infants was also examined to determine whether there is a relationship between the mother’s breast milk and infant gut bacterial populations. The main ethnic groups in NZ are the indigenous Māori (15%) and three major immigrant populations from the Pacific Islands (7%), Asia (12%) and Europe (74%) [16].

## 2. Materials and Methods

### 2.1. Study Design

The study protocol was approved by the NZ Human Disability and Ethics Committee (Application number 13/CEN/79/AM01) and was conducted in accordance with the Declaration of Helsinki. The study design has been described previously by Butts et al. [17]. Briefly, breast feeding women (aged 18–55 years) permanently living in the Manawatu–Wanganui region of the North Island of NZ were sought. One hundred and forty-six participants were screened, sixty-six participants who did not meet the recruitment criteria were excluded from the study, and the remaining eighty participants fulfilling the inclusion and exclusion criteria and who gave written consent were recruited into the study (Figure 1). The ethnicities of these women were Asian, Māori and Pacific Island, and Europeans. They each provided three breast milk samples (approximately 50 mL per sample) over a one-week period between six and eight weeks postpartum. These samples from each mother were immediately frozen in household freezers at −18 °C, brought to the laboratory within 12 h of collection of the last sample, thawed, pooled, aliquoted into smaller volumes, and then refrozen at −80 °C until analysis. In addition, fecal samples from the mother and the infant were also collected within the same week and frozen at −80 °C until analysis.

### 2.2. Microbiota Quantification

A 1 mL aliquot of each of the pooled breast milk samples from each individual was thawed to 5 °C prior to centrifugation at 7200 × *g* for 20 min to separate the fat, supernatant and pellet. A sterile spatula was used to remove the fat layer. The supernatant was removed leaving behind the pellet, which was re-suspended in 750 μL of bead-beating solution from the PowerLyzer PowerSoil DNA isolation kit (MO BIO Laboratories, Carlsbad, CA, USA). The re-suspended pellet and bead-beating solution were then transferred into a tube. A 0.25 g (±10%) portion of each fecal sample from mothers and infants was weighed into a sterile bead-beating tube. The DNA extraction for each sample was undertaken using the PowerLyzer PowerSoil DNA isolation kit (MO BIO Laboratories, Carlsbad, CA, USA) according to the manufacturer’s instructions. The bead-beating protocol was slightly modified, where a Disruptor Genie (Scientific Industries Inc., Bohemia, NY, USA) was used for 30 s followed by transferring the samples onto ice for 1 min; this process was repeated three times.

The PCR reactions were performed to target the V3–V4 hypervariable region of the 16s rRNA gene using the following primers [18]:

Forward: Bakt_341F-TCGTCGGCAGCGTCAGATGTGTATAAGAGACAGCCTACGGGNGGCWGCAG

and Reverse: Bakt_805R-GTCTCGTGGGCTCGGAGATGTGTATAAGAGACAGGACTACHVGGGTATCTAATCC.

The underlined sequence binds to the amplicon region of interest, whereas the remaining primer sequence binds to the Illumina index adaptors. The PCR conditions employed for breast milk and fecal bacterial DNA were slightly different. For breast milk samples, each PCR run consisted of 25 µL of HotStartTaq master mix (Qiagen, Melbourne, Australia), 11.5 µL of each of the forward and reverse primers and 2 µL of template DNA or sterile water (negative control), equaling 50 µL total volume. The PCR conditions were: initial denaturation of 95 °C for 15 min followed by 35 cycles of 30 s denaturation at 95 °C, 30 s annealing at 55 °C and 30 s extension at 72 °C, then lastly 5 min extension at 72 °C. For fecal samples, each PCR run consisted of 25 µL of HotStartTaq master mix (Qiagen), 12 µL of each of the forward and reverse primers and 1 µL of template DNA or sterile water (negative control) equaling 50 µL total volume. The PCR conditions were: initial denaturation of 95 °C for 15 min followed by 30 cycles of 30 s denaturation at 95 °C, 30 s annealing at 55 °C and 30 s extension at 72 °C, then lastly 5 min extension at 72 °C. The purification of PCR products was undertaken using a QIAquick PCR purification kit (Qiagen). The Qubit 2.0 Fluorometer (Life Technologies, Carlsbad, CA, USA) was used to quantify the PCR products before they were sent to the NZ Genomics Ltd. (Massey Genome Service, Palmerston North, NZ) for the second PCR step. The amplicons were library quality control checked, diluted and pooled prior to being sequenced on the Illumina MiSeq instrument (2 × 250 bp paired-end reads).

Quantitative Insights into Microbial Ecology (QIIME) software V1.8.0 was used to analyze the Illumina MiSeq sequencing data [19]. To assemble the forward and reverse reads into a continuous sequence, PANDASeq was used with parameters of at least 40 bp overlap, a minimum of 350 bp in length and a maximum of 500 bp in length [20]. The split_libraries.py script was run to enable assignment of appropriate sample names against each read. Chimeras were filtered from the sequences and the reads clustered into operational taxonomic units (OTUs) based on a 97% identity threshold value using USEARCH [21,22]. Alignment of the sequences was carried out using PyNAST [23] with reference to the Greengenes core reference database (version 13_8) [24]. Alpha rarefaction was calculated using the Phylogenetic Diversity, Chao1, Observed Species and Shannon diversity metrics to a rarefaction depth of 10,000 sequences. Beta diversity was determined using UniFrac distances as input and EMPeror to visualize 3D relationships [25,26].

### 2.3. Immune and Growth Factor Assays

The amounts of IgA, transforming growth factor (TGF)-β1 and -β2, insulin-like growth factor (IGF)-1, soluble cluster of differentiation 14 (CD14), interleukin (IL)-2, IL-4, IL-6, IL-7, IL-8, IL-10 and interferon (IFN)-γ were measured in the breast milk samples as supplied by the mothers. Two 1 mL aliquots of each of the pooled breast milk samples from each individual were thawed at room temperature prior to sodium taurocholate and whey extractions. An equal volume of 12 mmol/L sodium taurocholate (synthesized from cholic acid) was added to thawed milk samples, incubated for 30 min and centrifuged at 12,000× *g* for 10 min at room temperature. Intermediate fractions (between the fat layer and pellet) were used in Cytometric Bead Arrays, TGFβ2 ELISA and IgA ELISA assays. Whey was prepared by centrifuging thawed milk at 800× *g* for 10 min at 4 °C and the supernatant was collected. The whey fractions were used to measure TGFβ1 and IGF-1. Cytometric Bead Array was used for soluble human proteins, along with Flex sets for human soluble CD14, IL-2, IL-4, IL-6, IL-7, IL-8, IL-10 and IFN-γ according to the manufacturer’s instructions.

The dilutions used for each protein were: IgA 1/5000, TGFβ1 1/2, TGFβ2 and IGF-1 ELISA assays were undiluted, and then for the cytometric bead array, SCD14 was diluted 1/5000, whilst all other proteins were run undiluted. These dilutions ensured that the concentrations in the samples were within the concentration ranges of the standard curves. IgA, TGFβ1, TGFβ2 and IGF-1 in breast milk were quantified by using the respective ELISA kits according to the manufacturer’s instructions. IgA was determined using the Human IgA ELISA quantitation and ELISA starter accessory kit (E80-102 and E101, Bethyl laboratories, Montgomery, TX, USA), with a minimal detectable concentration of 7.8 ng/mL, an intra-assay precision of <5% and an inter-assay precision of <10%. TGFβ1 was determined using the LEGEND MAX™ Total TGFβ1 ELISA kit (Biolegend, San Diego, CA, USA, Cat No: 436707), with a minimal detectable concentration of 3.5 pg/mL, an intra-assay precision of <4% and an inter-assay precision of <7%. TGFβ2 was determined using a human TGFβ2 ELISA kit (Cusabio Life Science, Wuhan, China, Cat No: CSB-E09783h), with a minimal detectable concentration of 7.81 pg/mL, an intra-assay precision of <8% and an inter-assay precision of <10%. IGF-1 was determined using the Human IGF-1 ELISA kit (Cusabio Life Science, Wuhan, China, Cat No: CSB-E04580h), with a minimal detectable concentration of 1.95 ng/mL, an intra-assay precision of <8% and an inter-assay precision of <10%. IL-2, IL-4, IL-6, IL-7, IL-8, IL-10 and IFN-γ were determined (Super-X Plex™ cytokine assay, Antigenix America Inc., Melville, NY, USA), with minimal detectable concentrations of 3 pg/mL (IL-2), 1 pg/mL (IL-4), 5 pg/mL (IL-6), 5 pg/mL (IL-7), 1 pg/mL (IL-8), 2 pg/mL (IL-10) and 3 pg/mL (IFN-γ), an intra-assay precision of <10% and an inter-assay precision <20%.

### 2.4. Statistical Analysis

Summary statistics (means and standard errors) were calculated for each of the ethnic groups. The relative bacterial abundance in breast milk, the mother’s feces and the infant’s feces were skewed, so Kruskal–Wallis non-parametric analysis of variance (ANOVA) was applied. Only genera where the relative abundance was greater than 0.5% in at least one sample were analyzed. The immunology data were also skewed, so ethnic groups were compared using Kruskal–Wallis non-parametric ANOVA; where there was a significant difference between groups, multiple comparisons were done using the Conover-Iman method. In recognition of the number of variables being tested, the ANOVA *p*-values form each data set (immune-related proteins, maternal bacteria, infant bacteria, categorized by ethnicity, maternal body mass index (BMI) or delivery method) were adjusted using the Benjamini–Hochberg (BH) false discovery method. Kruskall–Wallis analyses were carried out using Genstat (version 17, VSNi Ltd., Hemel Hempstead, UK). The Benjamini–Hochberg adjustment was carried out using the *p*.adjust procedure in the stats package of R (The R Foundation for Statistical Computing, Vienna, Austria).

## 3. Results

### 3.1. Study Participants

The demographics and baseline characteristics of the study participants were reported earlier in Butts et al. [17] and Appendix A. From eighty participants, seventy eight completed the study—53 NZ European, 17 Māori and Pacific Island and 8 Asian women (Appendix A). When the participants were categorized according to the BMI classifications outlined by the Ministry of Health, NZ [27], 27 were normal weight, 31 overweight and 20 obese (Appendix A). At six weeks postpartum, many mothers will retain some of the weight they gained during pregnancy. When the participants were categorized according to the mode of delivery, 14 were by cesarean and 63 by vaginal delivery for the bacterial analyses, whereas there were 14 by cesarean and 64 by vaginal delivery for the immune modulatory proteins.

### 3.2. Microbiota

At the phylum level, no significant differences (*p* > 0.05) in bacterial populations were observed between ethnicities in breast milk and fecal samples of mothers (Figure 2). Similarly, there were no significant differences (*p* > 0.05) in infant fecal bacteria analyzed by maternal ethnicity. The bacterial abundance in breast milk, mother and infant feces at the genus level are shown in Table 1, Table 2 and Table 3. In breast milk, the most prevalent bacteria were *Ruminococcaceae*, *Bifidobacterium* and *Lachnospiraceae*. The most prevalent bacteria in the mother’s feces were *Ruminococcaceae*, *Bacteroides* and *Lachnospiraceae*, whereas, in the infant feces, *Bifidobacterium and Bacteroides* were the most prevalent bacteria.

The bacterial data were also analyzed based on BMI categories (normal, overweight and obese). At the phylum level, no significant differences (*p* > 0.05) in bacterial composition were observed in the mother’s breast milk and feces, or in the infant feces (Figure 3). Similarly, no significant differences (*p* > 0.05) were observed in bacterial composition at the genus level in the mother’s breast milk and feces, or in the infant feces (Appendix A).

The bacterial data were also analyzed based on mode of delivery (cesarean or vaginal). At the phylum level, there were no significant differences (*p* > 0.05) in bacterial composition of the mother’s breast milk and feces who have delivered their babies through cesarean section or vaginally (Figure 4). At the genus level, there were no significant differences (*p* > 0.05) in bacterial composition of the mother’s breast milk and feces who have delivered their babies through cesarean section or vaginally (Appendix A). Infants delivered vaginally had significantly higher numbers of *Collinsella* (*p* < 0.001) Appendix A).

### 3.3. Immune and Growth Factors

The immune modulatory proteins present in breast milk from the different ethnic groups are shown in Figure 5. The TGFβ1 and sCD14 concentrations were significantly higher in the breast milk from Māori and Pacific Island women compared with women from Asian and NZ European ethnicities (*p* < 0.05), whereas the IL-7 concentration was lower in the breast milk from Asian women compared to that of NZ European women (*p* < 0.05). Data analysis based on BMI categories showed that overweight women had significantly higher concentrations of TGFβ1 and TGFβ2 in their breast milk compared to those women who had normal body weight (Figure 6). No significant differences were found in the immune-related proteins within the milk of mothers who delivered their babies by cesarean section or vaginally (*p* > 0.05) (Appendix A).

## 4. Discussion

We examined the microbiota composition and immune modulatory proteins present in the breast milk of women from different ethnic backgrounds living in NZ. The microbiota in human milk is diverse, with the large inter-individual variation linked to geographical and lifestyle differences [28]. Studies have highlighted the importance of maternal diet in promoting infant health [29,30,31], and the vital role that bacteria and oligosaccharides in breast milk play in the development and functioning of the infant’s gut and immune system [13]. Breastfeeding is known to establish a diverse gut microbiota composition with a stable beneficial bacterial population in infants [32,33].

In the current study, we found no significant differences in bacteria abundances in the breast milk or fecal samples of mothers and infants when the data were compared by ethnicity, body mass and method of infant delivery. The mode of infant delivery is known to shape the structure of microbiota composition in infants. Previous studies have highlighted this by comparing the microbiota of infants delivered naturally through the vagina or by cesarean section [3,34]. In addition, the mode of delivery can have an impact on microbiota composition in maternal breast milk; cesarean section delivery was linked with reduced breast milk microbiota diversity and richness [7]. In the present study, when microbiota was analyzed according to the mode of delivery, the infants born through vaginal delivery had higher *Collinsella* genera in feces; although these differences were not evident in maternal breast milk or feces. An association between the abundance of gastrointestinal *Collinsella* spp. and body weight has been reported in earlier studies in overweight and obese pregnant women [35] and obese children [36].

The immune modulatory proteins present in breast milk contribute to the development of host defense mechanisms in the developing infant. They can have a range of impacts on infant health and their concentrations in breast milk are affected by maternal lifestyle factors, which are summarized in Appendix A. A balance of stimulatory and regulatory immune proteins is required to provide protection to the infant, whilst ensuring that inappropriate inflammation does not occur. Breast milk contains a number of factors that can impact mucosal immune function, including immune cells, antibodies, microbiota, oligosaccharides, cytokines, and soluble receptors. The concentrations of immune modulatory proteins found in the present study were highly variable. However, this has been reported previously in similar studies on breast milk [37,38,39]. This may be due to different methods of milk collection, including age of infant, time of milk collection, analytical method as well as population characteristics such as diet, gestation and exercise between and within studies [37,40]. In the present study, we limited this variability by collecting milk samples within a narrow infant age (6–8 weeks postpartum) and after the first feed of the day.

In the current study, we found that women of Māori and Pacific Island ethnicity had higher concentrations of TGFβ1 and sCD14 in their breast milk. IgA contributes to the exclusion of antigens in infants and its presence in breast milk is important while the infant’s own intestinal IgA production is not fully developed [41]. TGFβ has a role in IgA production, oral tolerance towards the infant’s own gut microbiota and preventing adverse immune reactions in infants [42], while sCD14 is associated with reduced risk of allergy and obesity [43]. Increased TGFβ1 levels in the breast milk of mothers have been observed in those with greater exposure to infectious diseases in childhood [44,45]. We also observed that there were higher levels of IL-7 cytokine in the breast milk of NZ European women when compared with the breast milk of Asian women. There is evidence that cytokines such as interleukins and IFN-γ cross the intestinal barrier where they communicate with cells and influence immune activity [46], and that regulatory cytokines such as IL-7 in breast milk are capable of crossing the intestinal barrier and inducing the development of the thymus of the infants and enhancing the production of T cells [47,48]. The importance of this observation in the context of the present study is not clear, as little is known about the influence of ethnicity and lifestyle factors on breast milk cytokines.

## 5. Conclusions

This is the first study to examine the microbiota composition and immune modulatory proteins in human breast milk from different ethnic populations in NZ. This study also explored the relationship between ethnicity, and maternal and infant fecal microbiota compositions, providing foundation data for further research on maternal and infant nutrition and the long-term health of people in NZ.

## Figures and Tables

**Figure 1 nutrients-12-01756-f001:**
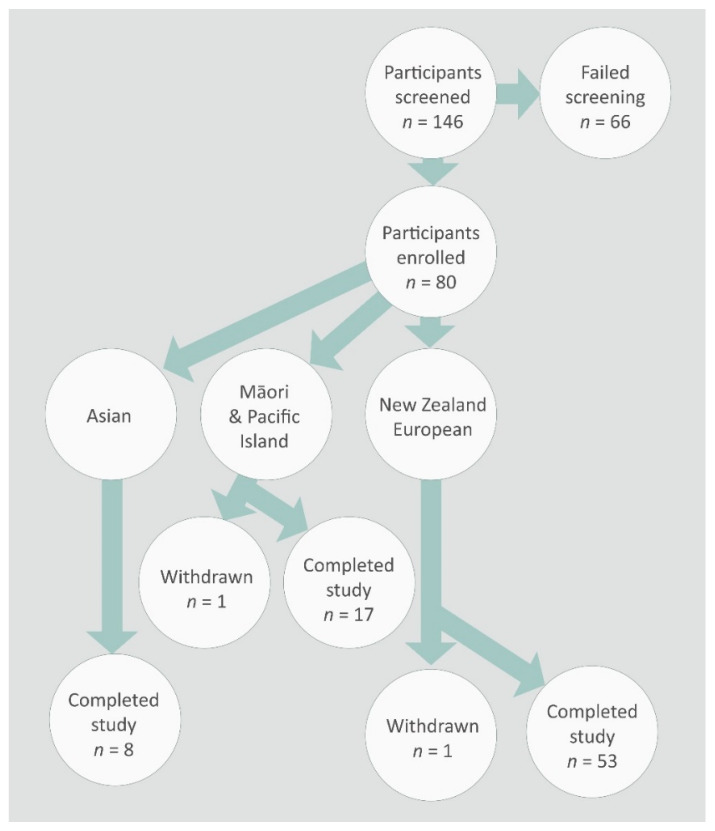
Flow chart of the participant recruitment into the study.

**Figure 2 nutrients-12-01756-f002:**
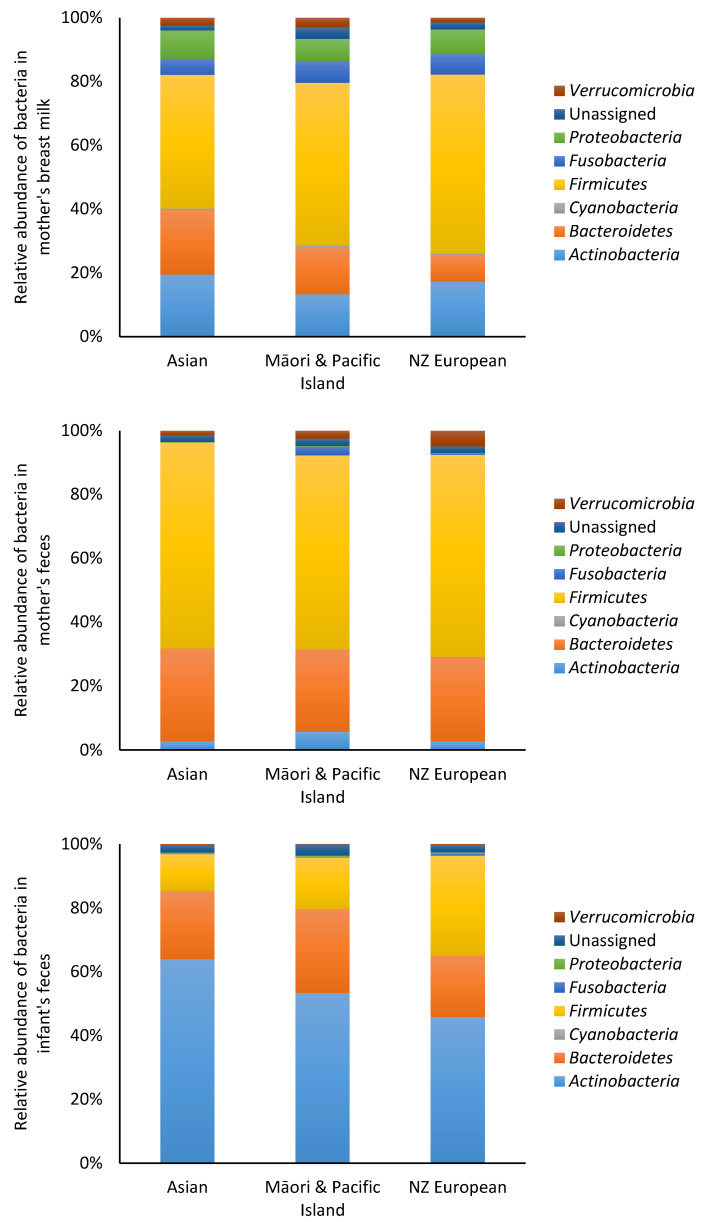
Relative abundance of bacteria at the phylum level in mother and infant. The data were categorized according to the mother’s ethnicity.

**Figure 3 nutrients-12-01756-f003:**
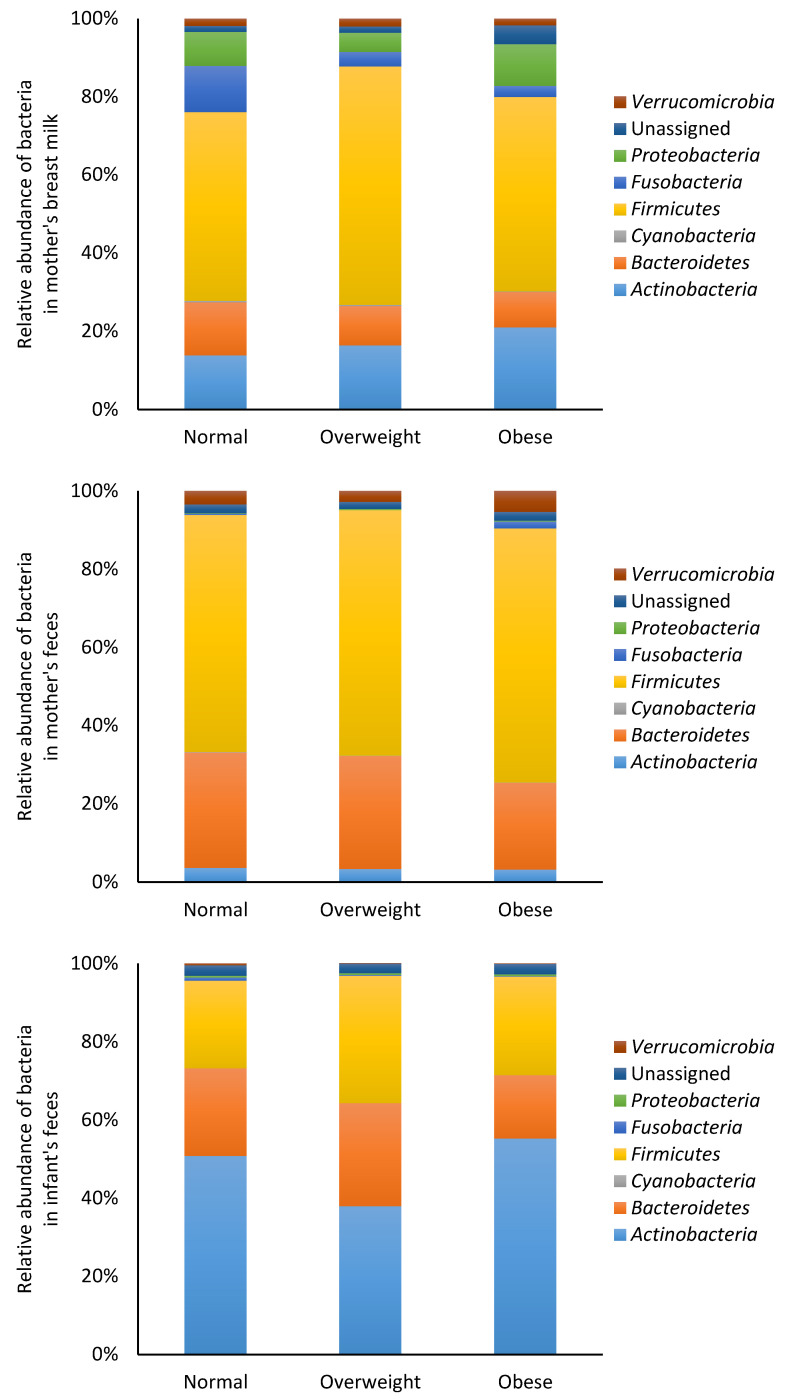
Relative abundance of bacteria at the phylum level in mother and infant. The data were categorized according to the mother’s body mass index.

**Figure 4 nutrients-12-01756-f004:**
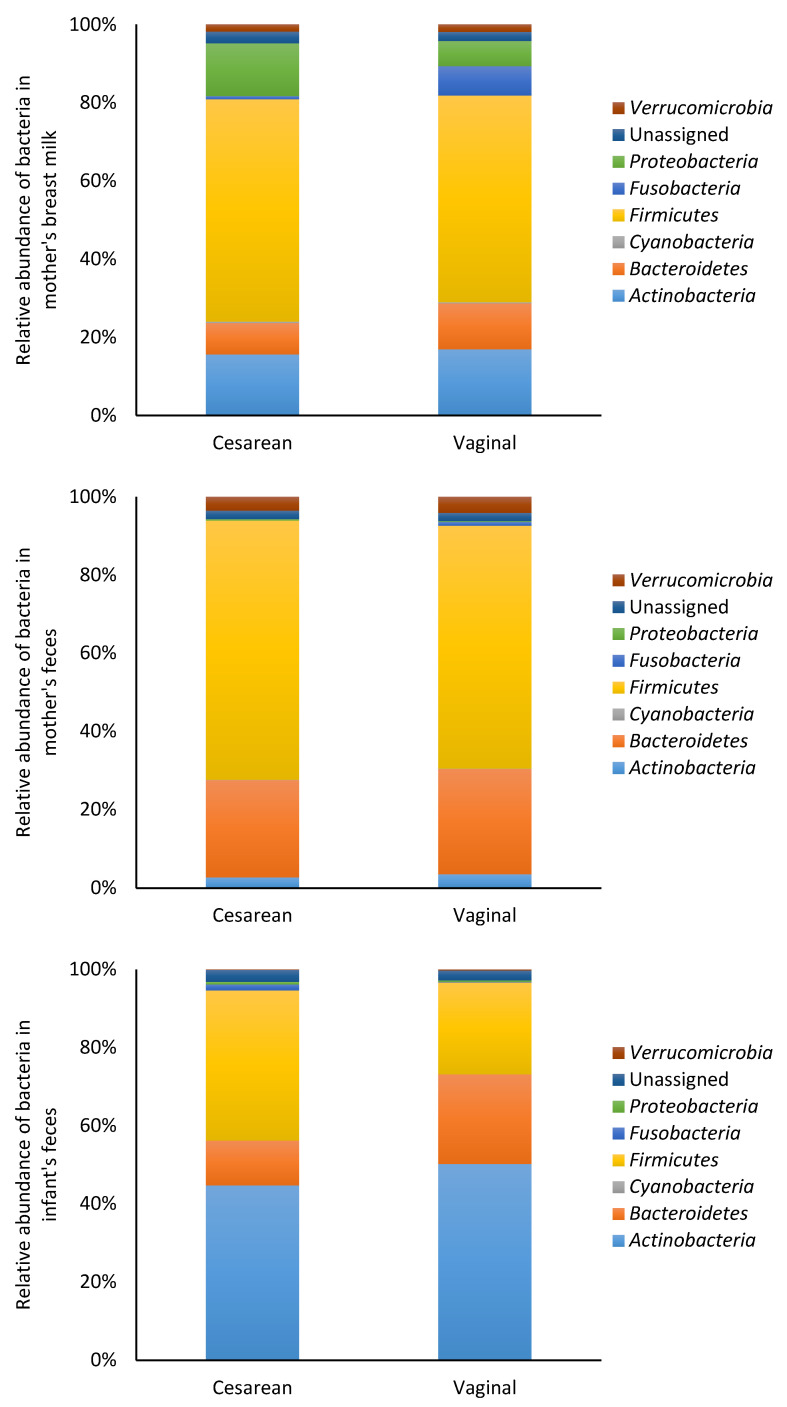
Relative abundance of bacteria at the phylum level in mother and infant. The data were categorized according to the mode of delivery.

**Figure 5 nutrients-12-01756-f005:**
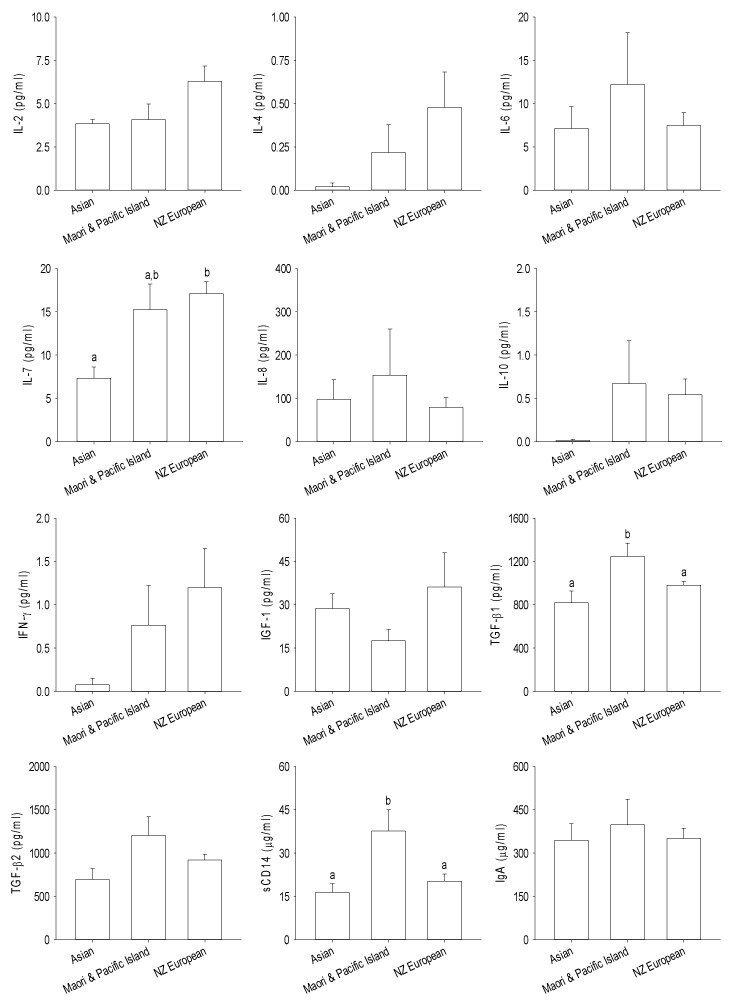
Immune-related proteins in the mother’s breast milk categorized according to the ethnicity. Data expressed as the mean ± standard error of the mean. Mean values with a different letter differ significantly, *p* < 0.05. IL—interleukin, IFN—interferon, IGF—insulin-like growth factor, TGF—transforming growth factor, sCD—soluble cluster of differentiation, and Ig—immunoglobulin.

**Figure 6 nutrients-12-01756-f006:**
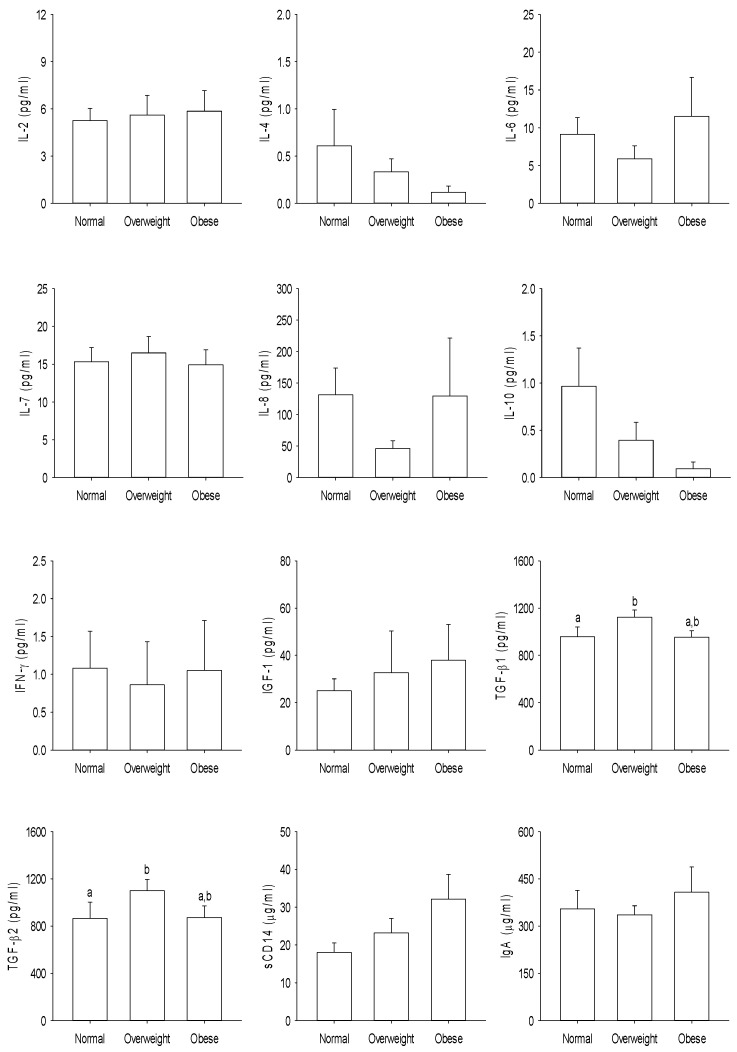
Immune-related proteins in the mother’s breast milk categorized according to the body mass index outlined by the Ministry of Health, New Zealand. Data expressed as the mean ± standard error of the mean. Mean values with a different letter differ significantly, *p* < 0.05. IL—interleukin, IFN—interferon, IGF—insulin-like growth factor, TGF—transforming growth factor, sCD—soluble cluster of differentiation, and Ig—immunoglobulin.

**Table 1 nutrients-12-01756-t001:** Relative abundance of bacteria at the genus level in breast milk categorized according to the mother’s ethnicity.

	Asian	Māori and Pacific Island	New Zealand European	Benjamini–Hochberg-Adjusted *p* Value
*Actinomyces*	0.48 ± 0.47	0.12 ± 0.10	0.46 ± 0.25	0.969
*Agrobacterium*	1.20 ± 0.95	0.09 ± 0.07	0.85 ± 0.64	0.858
*Akkermansia*	2.34 ± 1.14	3.01 ± 0.68	1.41 ± 0.29	0.376
*Anaerococcus*	0.15 ± 0.15	0.11 ± 0.07	0.26 ± 0.14	0.969
*Anaerostipes*	0	0.10 ± 0.10	0.03 ± 0.03	0.376
*Atopobium*	0.66 ± 0.65	0.54 ± 0.53	0.19 ± 0.11	0.980
*Bacteroidales*_S24_7	0.02 ± 0.02	0.12 ± 0.07	0.03 ± 0.01	0.773
*Bacteroides*	1.63 ± 0.80	3.88 ± 1.04	2.61 ± 0.50	0.748
*Barnesiellaceae*	0.04 ± 0.03	0.04 ± 0.03	0.05 ± 0.03	0.969
*Bifidobacteriaceae*	0.01 ± 0.01	0.25 ± 0.20	0.06 ± 0.04	0.969
*Bifidobacterium*	6.38 ± 2.26	6.08 ± 1.10	7.87 ± 1.01	0.897
*Blautia*	2.68 ± 0.90	3.97 ± 0.60	2.15 ± 0.27	0.376
*Brevundimonas*	0.01 ± 0.01	0.15 ± 0.14	0.21 ± 0.20	0.748
*Caulobacteraceae*	1.12 ± 0.95	0.88 ± 0.69	1.28 ± 0.51	0.531
*Christensenella*	0	0	0.04 ± 0.04	0.865
*Christensenellaceae*	0.39 ± 0.37	0.48 ± 0.15	0.21 ± 0.05	0.390
*Chryseobacterium*	0.06 ± 0.03	2.90 ± 2.81	2.39 ± 1.22	0.969
*Clostridiaceae*	0.70 ± 0.46	1.24 ± 0.35	0.93 ± 0.17	0.682
*Clostridiaceae*_Other	0.02 ± 0.02	0.24 ± 0.12	0.15 ± 0.05	0.969
*Clostridiales*	3.41 ± 1.38	4.38 ± 0.74	2.53 ± 0.36	0.402
*Clostridiales*_Other	0.89 ± 0.47	0.47 ± 0.11	0.48 ± 0.10	0.782
*Clostridium*	1.12 ± 0.74	2.00 ± 1.05	0.88 ± 0.25	0.460
*Collinsella*	0.92 ± 0.35	1.09 ± 0.23	1.23 ± 0.17	0.969
*Coprococcus*	2.13 ± 0.85	1.89 ± 0.38	1.73 ± 0.28	0.865
*Coriobacteriaceae*	0.56 ± 0.33	0.57 ± 0.17	0.25 ± 0.07	0.396
*Corynebacterium*	0.89 ± 0.34	2.92 ± 2.13	4.02 ± 1.20	0.969
*Dorea*	0.51 ± 0.37	1.00 ± 0.36	0.66 ± 0.26	0.376
*Elizabethkingia*	16.2 ± 10.6	0.03 ± 0.02	0.02 ± 0.01	0.403
*Enterobacteriaceae*	0.15 ± 0.15	0.04 ± 0.02	0.20 ± 0.13	0.969
*Epulopiscium*	0	0	0.02 ± 0.02	0.969
*Faecalibacterium*	0.52 ± 0.27	1.70 ± 0.28	0.89 ± 0.15	0.376
*Finegoldia*	0.03 ± 0.03	0.03 ± 0.03	0.16 ± 0.08	0.460
*Fusobacterium*	0.75 ± 0.45	0.61 ± 0.46	0.49 ± 0.19	0.624
*Lachnospira*	1.27 ± 0.63	1.45 ± 0.62	0.78 ± 0.17	0.518
*Lachnospiraceae*	4.71 ± 2.55	6.80 ± 0.93	5.05 ± 0.82	0.461
*Lachnospiraceae*_Other	0.14 ± 0.05	1.11 ± 0.37	0.29 ± 0.07	0.177
*Leucobacter*	0	0	0.26 ± 0.23	0.773
*Micrococcaceae*	0	0.01 ± 0.00	0.04 ± 0.04	0.461
*Micrococcus*	0.01 ± 0.01	0	0.07 ± 0.05	0.773
*Mogibacteriaceae*	0.16 ± 0.12	0	0.10 ± 0.07	0.714
*Odoribacter*	0	0	0.04 ± 0.04	0.461
*Oscillospira*	0.63 ± 0.46	0.65 ± 0.20	0.56 ± 0.14	0.599
*Parabacteroides*	0.26 ± 0.17	0.33 ± 0.12	0.29 ± 0.09	0.858
*Peptostreptococcus*	0.29 ± 0.19	0.04 ± 0.04	0.39 ± 0.14	0.402
*Porphyromonas*	0.17 ± 0.17	0	0.13 ± 0.09	0.682
*Prevotella*	0.01 ± 0.01	0.08 ± 0.03	0.27 ± 0.13	0.748
*Propionibacterium*	6.07 ± 5.15	0.35 ± 0.10	0.27 ± 0.09	0.390
*Pseudomonas*	0.28 ± 0.16	4.31 ± 3.04	0.96 ± 0.56	0.854
*Rikenellaceae*	1.92 ± 1.52	6.98 ± 1.92	2.01 ± 0.60	0.710
*Roseburia*	0.52 ± 0.28	0.56 ± 0.22	0.46 ± 0.12	0.969
*Rothia*	3.10 ± 1.78	0.83 ± 0.40	1.94 ± 0.76	0.969
*Ruminococcaceae*	10.7 ± 4.13	17.2 ± 2.05	10.2 ± 1.51	0.390
[*Ruminococcus*]	1.17 ± 0.80	1.32 ± 0.37	0.83 ± 0.30	0.376
*Ruminococcus*	2.05 ± 0.84	3.03 ± 0.68	1.49 ± 0.26	0.376
*Sphingobium*	0.11 ± 0.10	0.10 ± 0.05	0.83 ± 0.43	0.390
*Sphingomonas*	0.21 ± 0.09	0.41 ± 0.26	0.52 ± 0.15	0.969
*Streptophyta*	0.22 ± 0.21	0.10 ± 0.06	0.24 ± 0.09	0.748
Unassigned	1.51 ± 0.30	3.62 ± 1.24	2.20 ± 0.47	0.682
*Veillonella*	0.07 ± 0.05	0.02 ± 0.01	0.27 ± 0.08	

Data expressed as the mean ± standard error of the mean. Some taxonomy could not be identified at the genus level. Square brackets around a genus name indicate a candidate genus.

**Table 2 nutrients-12-01756-t002:** Relative abundance of bacteria at the genus level in the mother’s feces categorized according to the ethnicity.

	Asian	Māori and Pacific Island	New Zealand European	Benjamini–Hochberg-Adjusted *p* Value
*Actinomyces*	0	0	0.01 ± 0.01	0.471
*Agrobacterium*	0	0	0.04 ± 0.04	0.499
*Akkermansia*	1.42 ± 0.84	2.42 ± 0.78	3.17 ± 1.02	0.453
*Anaerococcus*	0	0.01 ± 0.01	0.03 ± 0.03	0.257
*Anaerostipes*	0.07 ± 0.05	0.08 ± 0.03	0.08 ± 0.04	0.453
*Bacteroidales*	0	0.33 ± 0.33	0.10 ± 0.10	0.894
*Bacteroidales*_S24_7	0	0.09 ± 0.08	0.83 ± 0.43	0.629
*Bacteroides*	20.5 ± 7.53	16.6 ± 4.30	17.4 ± 2.25	0.894
*Barnesiellaceae*	0.38 ± 0.19	1.19 ± 0.51	0.96 ± 0.19	0.931
*Bifidobacterium*	2.22 ± 0.88	3.65 ± 1.42	1.57 ± 0.31	0.240
*Blautia*	1.94 ± 0.46	3.25 ± 1.11	2.73 ± 0.33	0.748
*Butyricimonas*	0.10 ± 0.04	0.08 ± 0.04	0.13 ± 0.05	0.629
*Caulobacteraceae*	0.07 ± 0.07	0	0.31 ± 0.09	0.240
*Christensenellaceae*	0.50 ± 0.47	0.48 ± 0.16	0.55 ± 0.09	0.240
*Clostridiaceae*	0.16 ± 0.08	0.60 ± 0.21	0.34 ± 0.07	0.240
*Clostridiaceae*_Other	0.03 ± 0.02	0.11 ± 0.07	0.03 ± 0.01	0.894
*Clostridiales*	4.90 ± 1.61	3.25 ± 0.89	5.49 ± 0.54	0.240
*Clostridiales*_Other	0.85 ± 0.25	0.49 ± 0.10	0.98 ± 0.16	0.453
*Clostridium*	0.40 ± 0.13	0.22 ± 0.04	0.34 ± 0.03	0.407
*Collinsella*	0.36 ± 0.09	1.44 ± 0.53	0.64 ± 0.10	0.352
*Coprococcus*	3.10 ± 0.45	2.63 ± 0.60	2.81 ± 0.42	0.580
*Coriobacteriaceae*	0.11 ± 0.06	0.32 ± 0.07	0.37 ± 0.08	0.294
*Dorea*	0.45 ± 0.14	1.35 ± 0.45	0.58 ± 0.10	0.240
*Eggerthella*	0.03 ± 0.01	0.04 ± 0.02	0.06 ± 0.02	0.986
*Enterobacteriaceae*	0	0.01 ± 0.01	0.10 ± 0.04	0.453
*Epulopiscium*	0	0	0.11 ± 0.11	0.580
*Faecalibacterium*	2.39 ± 0.87	1.69 ± 0.33	1.44 ± 0.28	0.352
*Finegoldia*	0.21 ± 0.21	0.02 ± 0.02	0.20 ± 0.04	0.453
*Fusobacterium*	0	2.71 ± 2.69	0.02± 0.01	0.629
*Lachnospira*	2.62 ± 0.96	2.01 ± 1.00	1.45 ± 0.38	0.320
*Lachnospiraceae*	7.24 ± 1.61	6.20 ± 1.15	6.05 ± 0.88	0.664
*Lachnospiraceae*_Other	0.49 ± 0.14	0.75 ± 0.13	0.42 ± 0.07	0.240
*Mogibacteriaceae*	0.04 ± 0.04	0.10 ± 0.05	0.05 ± 0.01	0.748
*Odoribacter*	0.10 ± 0.06	0.19 ± 0.09	0.19 ± 0.04	0.663
*Oscillospira*	0.47 ± 0.17	0.84 ± 0.17	0.88 ± 0.20	0.664
*Parabacteroides*	1.03 ± 0.45	1.71 ± 0.53	1.67 ± 0.39	0.926
*Paraprevotella*	0	0.49 ± 0.29	0.46 ± 0.37	0.240
*Porphyromonas*	0.02 ± 0.01	0	0.03 ± 0.02	0.472
*Prevotella*	5.58 ± 5.57	2.38 ± 2.16	0.826 ± 0.26	0.748
*Rikenellaceae*	1.05 ± 0.34	2.64 ± 0.86	3.51 ± 0.58	0.626
*Rikenellaceae_*Other	0	0	0.02 ± 0.01	0.240
*Roseburia*	2.13 ± 1.75	0.75 ± 0.40	0.40 ± 0.12	0.664
*Ruminococcaceae*	26.1 ± 6.54	26.4 ± 3.88	19.1 ± 2.70	0.406
[*Ruminococcus*]	0.41 ± 0.14	2.38 ± 1.52	0.86 ± 0.19	0.294
*Ruminococcus*	4.05 ± 1.71	3.93 ± 1.09	3.26 ± 0.61	0.627
*Sphingomonas*	0.01 ± 0.01	0	0.03 ± 0.01	0.240
*Sutterella*	0.12 ± 0.06	0.02 ± 0.01	0.09 ± 0.05	0.453
Unassigned	2.06 ± 0.21	2.29 ± 0.27	2.11 ± 0.17	0.580
*Veillonella*	0.95 ± 0.95	0.09 ± 0.09	2.25 ± 0.63	0.254

Data expressed as the mean ± standard error of the mean. Some taxonomy could not be identified at the genus level. Square brackets around a genus name indicate a candidate genus.

**Table 3 nutrients-12-01756-t003:** Relative abundance of bacteria at the genus level in the infant’s feces categorized according to the mother’s ethnicity.

	Asian	Māori and Pacific Island	New Zealand European	Benjamini–Hochberg-Adjusted *p* Value
*Actinomyces*	0.01 ± 0.006	0.02 ± 0.02	1.31 ± 0.86	0.706
*Agrobacterium*	0	0.005 ± 0.005	0.07 ± 0.05	0.706
*Akkermansia*	0.39 ± 0.37	0.07 ± 0.05	0.01 ± 0.01	0.706
*Anaerotruncus*	0	0	0.02 ± 0.02	0.706
*Bacteroides*	20.1 ± 10.8	24.12± 7.52	16.01 ± 3.57	0.706
*Bifidobacterium*	60.1 ± 12.4	50.4 ± 8.270	41.2 ± 4.94	0.706
*Blautia*	0.23 ± 0.19	0.12 ± 0.06	0.48 ± 0.24	0.965
*Caulobacteraceae*	0	0	1.74 ± 1.16	0.706
*Clostridiaceae*	6.13 ± 6.00	2.80 ± 2.69	3.14 ± 1.84	0.706
*Clostridiaceae*_Other	0	0.01 ± 0.01	0.02 ± 0.02	0.936
*Clostridiales*	0.43 ± 0.37	0.12 ± 0.08	0.45 ± 0.31	0.706
*Clostridiales*_Other	0.02 ± 0.01	0.02 ± 0.01	0.08 ± 0.04	0.706
*Clostridium*	0.18 ± 0.09	6.21 ± 5.78	14.3 ± 4.16	0.706
*Collinsella*	3.70 ± 3.42	2.47 ± 1.05	2.49 ± 1.08	0.706
*Coprococcus*	0.32 ± 0.28	0.08 ± 0.05	0.23 ± 0.10	0.706
*Coriobacteriaceae*	0.05 ± 0.03	0.23 ± 0.16	0.43 ± 0.36	0.706
*Dorea*	0.08 ± 0.06	0.07 ± 0.07	0.63 ± 0.49	0.719
*Eggerthella*	0.05 ± 0.04	0.12 ± 0.08	0.13 ± 0.06	0.706
*Enterobacteriaceae*	0.36 ± 0.30	0.35 ± 0.13	0.33 ± 0.07	0.706
*Faecalibacterium*	0.22 ± 0.19	0.09 ± 0.07	0.03 ± 0.01	0.706
*Finegoldia*	0.01 ± 0.01	0	0.04 ± 0.03	0.706
*Fusobacterium*	0.03 ± 0.03	0.03 ± 0.02	0.34 ± 0.33	0.706
*Lachnospiraceae*	1.11 ± 0.82	2.32 ± 1.65	0.80 ± 0.31	0.706
*Lachnospiraceae*_Other	0.02 ± 0.02	0.06 ± 0.02	0.68 ± 0.63	0.724
*Megasphaera*	0	0.36 ± 0.26	0.04 ± 0.04	0.706
*Odoribacter*	0	0	0.07 ± 0.05	0.706
*Oscillospira*	0.31 ± 0.27	0.01 ± 0.01	0.26 ± 0.13	0.706
*Parabacteroides*	0.22 ± 0.14	2.05 ± 1.07	2.57 ± 1.37	0.936
*Peptostreptococcus*	0.03 ± 0.03	0.13 ± 0.13	0.03 ± 0.03	0.936
*Porphyromonas*	0	0	0.06 ± 0.05	0.706
*Prevotella*	0.01 ± 0.01	0.03 ± 0.02	0.12 ± 0.08	0.936
*Pseudoramibacter*	0	0.22 ± 0.22	0.03 ± 0.03	0.833
*Rikenellaceae*	0.83 ± 0.83	0.01 ± 0.00	0.40 ± 0.18	0.706
*Ruminococcaceae*	0.92 ± 0.66	1.04 ± 0.61	1.39 ± 0.95	0.706
[*Ruminococcus*]	0.79 ± 0.73	1.85 ± 1.15	3.72 ± 1.83	0.706
*Ruminococcus*	0.51 ± 0.43	0.25 ± 0.18	0.09 ± 0.02	0.706
*Sphingomonas*	0.01 ± 0.01	0	0.27 ± 0.26	0.706
Unassigned	2.25 ± 0.60	3.56 ± 0.91	2.40 ± 0.26	0.706
*Veillonella*	0.01 ± 0.01	0.03 ± 0.01	0.54 ± 0.20	0.706

Data expressed as the mean ± standard error of the mean. Some taxonomy could not be identified at the genus level. Square brackets around a genus name indicate a candidate genus.

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
