# Peer review of "Microbiota Composition of Breast Milk from Women of Different Ethnicity from the Manawatu—Wanganui Region of New Zealand"

_nutrients, 2020, doi:10.3390/nu12061756_

Round 1

Reviewer 1 Report

This manuscript evaluated the effect of ethnicity, obesity and mode of delivery on the microbiota composition and the immune protein concentrations in human milk from women living in NZ. This research is interesting and novel, however, more statistical analysis should be performed to evaluate the effect of obesity and mode of delivery on the immune protein concentrations. A better description of the different groups of women in NZ would improve and help to understand the final data.

Introduction. 

The demographic characteristics of the different women ethnic groups living in New Zealand should be included in a paragraph (after line 74).

As the authors evaluated the effect of mode of delivery and obesity on the microbiome and immune components in breast milk, a brief overview of the literature on these aspects should be included in the intro 

Materials and Methods.

Please, specify if the TGF-beta 1 and 2 results are from activated breast milk samples (by acidification?)

Results.

The authors should include a table to describe the group of Asian, Maori & Pacific Island and Europeans: what are the difference in demographic characteristics of these women? Please, include a table with the number of women (n) who were "obese" vs "non-obese", cesarian vs vaginal, postnatal age, exclusively breastfeeding. 

Discussion.

Line 248. Authors could describe the Collinsella genera: Actinobacteria phylum includes Bifidobacterium and Collinsella, which are generally the oligosaccharide fermenters present in the distal gut. 
https://www.sciencedirect.com/science/article/pii/B9780128114407000119

Authors could also perform linear regression on level of breast milk immune component in function of the immune component level from infant’s feces.

I highly recommend doing statistical analysis between the group vaginal vs cesarian for each measured immune component in different biological samples (mother’s milk, mother’s feces, infant’ feces).

Line 267. The authors may could to add that TGF‐β can control excessive inflammatory responses by T‐cells and stimulate IgA production.
TGF-β1 in human milk may prevent adverse immunologic reactions in infants.
(Ref: Ohtsuka, Y.; Sanderson, I.R.Curr. Opin. Gastroenterol. 2000, 16, 541–545)
Deficiency in TGF-β1 or its receptor has been implicated in the development of inflammatory disease

Figure 5-6. It would be more appropriate to show individual mothers on each figure (points) for each group to see the difference between mothers.

Please, specify the statistical analysis used and the number of samples (n) in each group in the title on figure 5 and 6
What are the differences of characteristics (weight, diet, active lifestyle, mode of delivery, lactation time) between these ethnicities?

All tables: one significant number should be include for SD. Example:
0.48 +- 0.47 —> 0.5 +- 0.5
p-value: 2 significant numbers must be include: Example:
p = 0.969 —> p = 0.97

Author Response

Introduction. 

The demographic characteristics of the different women ethnic groups living in New Zealand should be included in a paragraph (after line 74).

Thank you for the suggestion and this has now been included in the manuscript.

As the authors evaluated the effect of mode of delivery and obesity on the microbiome and immune components in breast milk, a brief overview of the literature on these aspects should be included in the intro

We have expanded the Introduction to include more detail on this topic.

Materials and Methods.

Please, specify if the TGF-beta 1 and 2 results are from activated breast milk samples (by acidification?)

The breast milk samples supplied by the mothers were thawed and treated as described in the manuscript. Activating with acid may have affected the analyses of the selected proteins.

Results.

The authors should include a table to describe the group of Asian, Maori & Pacific Island and Europeans: what are the difference in demographic characteristics of these women? Please, include a table with the number of women (n) who were "obese" vs "non-obese", cesarian vs vaginal, postnatal age, exclusively breastfeeding. 

The demographics and baseline characteristics of the study participants were previously reported in the accompanying supplementary tables of Butts et al. [1]. We have now included them in the Supplementary Tables S1-S4 in the present manuscript for completeness.

The participants were selected based on their practising exclusive breast feeding. See “Breastfeeding had to be exclusive or mixed with no more than two formula feeds a day or water or medication, and the mothers were asked to record exactly what method of feeding they used.” in Butts et al. [1]. This was stringently adhered to.

Discussion.

Line 248. Authors could describe the Collinsella genera: Actinobacteria phylum includes Bifidobacterium and Collinsella, which are generally the oligosaccharide fermenters present in the distal gut. 
https://www.sciencedirect.com/science/article/pii/B9780128114407000119

Thank you for this suggestion for improving the manuscript. We have inserted the sentence “An association between the abundance of gastrointestinal Collinsella spp. and body weight has been reported in earlier studies in overweight and obese pregnant women [2] and obese children [3].”

Authors could also perform linear regression on level of breast milk immune component in function of the immune component level from infant’s feces.

We have not measured these immune proteins in the infant’s or mother’s faeces as our focus was primarily on breast milk composition in New Zealand mothers from different ethnicities.

I highly recommend doing statistical analysis between the group vaginal vs cesarian for each measured immune component in different biological samples (mother’s milk, mother’s feces, infant’ feces).

This information is presented in the submitted Supplementary file in Figure S1.

Line 267. The authors may could to add that TGF‐β can control excessive inflammatory responses by T‐cells and stimulate IgA production.
TGF-β1 in human milk may prevent adverse immunologic reactions in infants. 
(Ref: Ohtsuka, Y.; Sanderson, I.R.Curr. Opin. Gastroenterol. 2000, 16, 541–545)
Deficiency in TGF-β1 or its receptor has been implicated in the development of inflammatory disease

Thank you for this suggestion and this reference has now been included.

Figure 5-6. It would be more appropriate to show individual mothers on each figure (points) for each group to see the difference between mothers.

We prefer to present the data as currently shown which is as the mean ± standard error as this provides important information on the variation in the study population.

Please, specify the statistical analysis used and the number of samples (n) in each group in the title on figure 5 and 6.

This information is provided in the Materials and Methods and Results sections.

What are the differences of characteristics (weight, diet, active lifestyle, mode of delivery, lactation time) between these ethnicities?

These characteristics were not investigated in the present study.

All tables: one significant number should be include for SD. Example:
0.48 +- 0.47 —> 0.5 +- 0.5
p-value: 2 significant numbers must be include: Example:
p = 0.969 —> p = 0.97

Thank you for this suggestion, however, we noted in doing this there was a loss of accuracy in the data particularly where there is a wide variation in the number of significant fingers between the treatment groups such as the bacterial abundance. The data is presented as 3 significant figures through out the manuscript and supplementary file for consistency.

Reviewer 2 Report

Very good paper, I was delighted to review it.

I have two questions.

METHOD

84 Briefly, breast feeding

85 women (aged 18-55 years) permanently living in the Manawatu-Wanganui region of the North Island

86 of NZ were sought.

Question 1. Is this age range correct?, the lower limit and the upper limit both look too high

RESULTS

Question 2. Apparently, there are striking differences of microbiota and immue factors between obese mothers and mothers with normal BMI. These differences, excluding mothers with overweight, are worth to be explored.

Author Response

Very good paper, I was delighted to review it.

I have two questions.

METHOD

84 Briefly, breast feeding

85 women (aged 18-55 years) permanently living in the Manawatu-Wanganui region of the North Island

86 of NZ were sought.

Question 1. Is this age range correct?, the lower limit and the upper limit both look too high.

Yes, this is the correct range that we used to recruit the participants. The age range of the recruited participants was 19-42 years. This information is now included in the supplementary file (Table S1).

RESULTS

Question 2. Apparently, there are striking differences of microbiota and immune factors between obese mothers and mothers with normal BMI. These differences, excluding mothers with overweight, are worth to be explored.

Thank you for this suggestion, we have included some more information in the Discussion.

Round 2

Reviewer 1 Report

The authors added or corrected appropriately the suggested modifications, except for the visualization of the figures (individual values). I don't have more suggestions for the authors.